# Characterization of the Aerobic Anoxygenic Phototrophic Bacterium *Sphingomonas* sp. AAP5

**DOI:** 10.3390/microorganisms9040768

**Published:** 2021-04-06

**Authors:** Karel Kopejtka, Yonghui Zeng, David Kaftan, Vadim Selyanin, Zdenko Gardian, Jürgen Tomasch, Ruben Sommaruga, Michal Koblížek

**Affiliations:** 1Centre Algatech, Institute of Microbiology, Czech Academy of Sciences, 379 81 Třeboň, Czech Republic; kopejk00@alga.cz (K.K.); marinezeng@gmail.com (Y.Z.); kaftan@alga.cz (D.K.); selyaninvv@gmail.com (V.S.); 2Department of Plant and Environmental Sciences, University of Copenhagen, Thorvaldsensvej 40, 1871 Frederiksberg C, Denmark; 3Faculty of Science, University of South Bohemia, 370 05 České Budějovice, Czech Republic; zdenogardian@gmail.com; 4Institute of Parasitology, Biology Centre, Czech Academy of Sciences, 370 05 České Budějovice, Czech Republic; 5Research Group Microbial Communication, Technical University of Braunschweig, 38106 Braunschweig, Germany; Juergen.Tomasch@helmholtz-hzi.de; 6Laboratory of Aquatic Photobiology and Plankton Ecology, Department of Ecology, University of Innsbruck, 6020 Innsbruck, Austria; Ruben.Sommaruga@uibk.ac.at

**Keywords:** aerobic anoxygenic phototrophic bacteria, bacteriochlorophyll a, photosynthesis genes, rhodopsin, Sphingomonadaceae

## Abstract

An aerobic, yellow-pigmented, bacteriochlorophyll *a*-producing strain, designated AAP5 (=DSM 111157=CCUG 74776), was isolated from the alpine lake Gossenköllesee located in the Tyrolean Alps, Austria. Here, we report its description and polyphasic characterization. Phylogenetic analysis of the 16S rRNA gene showed that strain AAP5 belongs to the bacterial genus *Sphingomonas* and has the highest pairwise 16S rRNA gene sequence similarity with *Sphingomonas glacialis* (98.3%), *Sphingomonas psychrolutea* (96.8%), and *Sphingomonas melonis* (96.5%). Its genomic DNA G + C content is 65.9%. Further, in silico DNA-DNA hybridization and calculation of the average nucleotide identity speaks for the close phylogenetic relationship of AAP5 and *Sphingomonas glacialis*. The high percentage (76.2%) of shared orthologous gene clusters between strain AAP5 and *Sphingomonas paucimobilis* NCTC 11030^T^, the type species of the genus, supports the classification of the two strains into the same genus. Strain AAP5 was found to contain C_18:1_*ω*7*c* (64.6%) as a predominant fatty acid (>10%) and the polar lipid profile contained phosphatidylglycerol, diphosphatidylglycerol, phosphatidylethanolamine, sphingoglycolipid, six unidentified glycolipids, one unidentified phospholipid, and two unidentified lipids. The main respiratory quinone was ubiquinone-10. Strain AAP5 is a facultative photoheterotroph containing type-2 photosynthetic reaction centers and, in addition, contains a xathorhodopsin gene. No CO_2_-fixation pathways were found.

## 1. Introduction

The genus *Sphingomonas* (Alphaproteobacteria) was originally proposed by Yabuuchi and coworkers [1] as a genus accommodating Gram-negative, strictly aerobic, non-sporulating, non-motile or motile, non-fermenting, chemoheterotrophic bacteria [2,3]. Later, this genus was divided into four genera and genus *Sphingomonas* was redefined in *sensu stricto* [4]. Over time, several photoheterotrophic representatives of *Sphingomonas* were cultivated [5,6,7,8]. *Sphingomonas* are found in a wide range of environmental niches, such as soils [6,9,10], fresh and marine waters [11,12,13], plants [14,15], airborne dust [16,17], and clinical samples [1,18,19]. Some representatives of *Sphingomonas* have a potential for biotechnological applications [20,21,22].

We previously isolated a novel *Sphingomonas* sp. strain designated AAP5 from the alpine lake Gossenköllesee located in the Tyrolean Alps, Austria. This aerobic yellow-pigmented strain contains bacteriochlorophyll-containing reaction centers [23].

Anoxygenic photosynthesis is relatively common among members of the order Sphingomonadales. Indeed, the first cultured aerobic anoxygenic phototrophic (AAP) bacterium was *Erythrobacter longus* isolated from the Bay of Tokyo [24]. Many AAP species have been cultured from freshwater, namely from the genera *Porphyrobacter*, *Erythromicrobium*, *Erythromonas, Sandarakinorhabdus*, and *Blastomonas* [25]. However, the unique feature of the strain AAP5 is that, together with genes for bacterial reaction center it contains also gene for another light harvesting protein xanthorhodopsin [23].

Here, we report the detailed phenotypic, phylogenetic, genomic, physiological, and biochemical characterization of the AAP5 strain. Furthermore, we compared it with its closest relative, *Sphingomonas glacialis*, and with the type species of the genus, *S. paucimobilis*.

## 2. Materials and Methods

*Sampling site and strain isolation.* The sampling was conducted in the clear alpine lake Gossenköllesee, Tyrolean Alps, Austria in September 2012. The lake is situated in a siliceous catchment area at 2417 m above sea level (47.2298° N, 11.0140° E). Details on sampling site and sampling procedure were described previously [12].

1 µl of the lake water sample was diluted into 100 µl of sterile half-strength R2A medium, and the dilution spread onto half-strength standard R2A agar plates (DSMZ medium 830). The plates were incubated aerobically at 25 °C under 12-h-light/dark cycles until colonies were visible, which were then screened for the presence of bacteriochlorophyll *a* (BChl *a*) using an infrared (IR) imaging system [26]. IR positive colonies were repeatedly streaked onto new agar plates until pure cultures were obtained.

*Cultivation conditions.* For all analyses conducted, the strain was grown either on R2A solid medium (DSMZ medium 830) or in R2A liquid medium. Cultures were incubated aerobically in 100 mL of the liquid medium in 250-mL flasks with cotton plugs on an orbital shaker (150 RPM). Illumination was provided by a bank of Dulux L 55W/865 luminescent tubes (Osram, Germany, spectral temperature of 6500 K) delivering ca. 100 µmol photons m^−2^ s^−1^. Unless stated otherwise, the cultures were grown under 12-h dark/12-h light regime and at 22°C. The growth of cultures was followed by turbidity measurements at 650 nm.

*Microscopy.* Samples for epifluorescence microscopy were diluted 1000-fold in a sterile medium, fixed with sterile-filtered formalin to a final concentration of 1%, filtered onto white polycarbonate filters (Nuclepore, pore size 0.2 µm, diameter 25 mm, Whatman) and stained with 4′,6-diamidino-2-phenylindole at final concentration of 1 mg l^−1^ [27]. The cells were visualized under UV/blue excitation emission, and the autofluorescence of BChl *a* was visualized under white light/IR emission, using a Zeiss Axio Imager.D2 microscope equipped with a Plan-Apochromat 63x/1.46 Oil Corr objective, a Hamamatsu EMCCD camera C9100-02Min and Collibri2 LED illumination, as described previously [28].


Samples for transmission electron microscopy (TEM) and scanning electron microscopy (SEM) were fixed with 2.5% glutaraldehyde in 0.1 M phosphate buffer (pH = 7.2) for 2 days at 4°C. TEM samples were post-fixed in osmium tetroxide for 2 h, at 4°C, washed, dehydrated through an acetone series and embedded in Spurr’s resin. A series of ultrathin sections were cut using a Leica UCT ultramicrotome (Leica Microsystems), counterstained with uranyl acetate and lead citrate, then examined in a JEOL TEM 1010 operated at 80 kV. SEM samples were dehydrated through an acetone series and dried by means of a critical point dryer CPD 2 (Pelco TM). Dry samples were attached to an aluminum target by means of carbon tape, coated with gold using a sputter coater E5100 (Polaron Equipment Ltd.) and examined with JEOL SEM JSM 7401F. Images were digitally recorded for the determination of morphological parameters.


Samples for atomic force microscopy (AFM) were resuspended in a buffer containing 20 mM Tris-HCl pH 8.0, 50 mM NaCl and adsorbed onto a clean glass coverslip functionalized by Corning™ Cell-Tak Cell and Tissue Adhesive (Corning Inc, USA). Cells were imaged by NanoWizard4 BioAFM (Bruker, USA) atop of an inverted optical microscope (IX73P2F, Olympus, Japan) placed on active vibration isolation system (Halcyonics) inside a custom-made acoustic enclosure. Cells were imaged with lever 3 of qp-BioAC AFM probe (Nanosensors, Switzerland). Cantilevers were calibrated using thermal noise, non-contact Sader method [29] providing resonant frequency f_o_ = 8.94 kHz and spring constant k = 0.076 N m^−1^. Quantitative Imaging maps (5 × 5 µm^2^, 10 × 10 µm^2^) were recorded with resolution of 256 × 256 pixels^2^. The force-distance curves recorded over the sample’s surface were baseline corrected and the vertical tip position was estimated by fitting the position of the contact point. The Young’s Modulus was calculated by fitting the processed curves using the Herz/Sneddon model [30,31] according to Rico and coworkers [32].

*Phylogenetic analyses.* The 16S rRNA gene sequence (GenBank accession number MW410774) of strain AAP5 was retrieved from its genome sequence (GenBank accession number GCA_004354345.1). Reference sequences were obtained either from the SILVA database [33] or NCBI GenBank (May 2020), and aligned using ClustalW [34]. Ambiguously aligned regions and gaps were manually excluded from further analysis. The 16S rRNA phylogenetic tree was computed using both neighbor-joining (NJ) [35] and maximum likelihood (ML) [36] algorithms included in the MEGA 6.06 software [37]. The Tamura-Nei model [38] was used for inferring the NJ tree. The ML tree was constructed using GTR nucleotide substitution model [39]. A uniform rate of nucleotide substitution was used. For the PufLM concatenated tree, amino acid sequences were retrieved from GenBank™ (December 2020) and aligned using ClustalX version 2.1. Sites containing gaps and ambiguously aligned regions were manually excluded. Amino acid sequence alignments of PufL and PufM were concatenated with Geneious version 8.1.2 (Biomatters Ltd.). Phylogenomic trees were inferred by MEGA 6.0 software using the ML algorithm with LG model [40] and 1000 bootstrap replicates.

In silico DNA-DNA hybridization (iDDH), average nucleotide identity (ANI), DNA base composition analysis, and orthologous gene cluster analysis. Genome sequences of AAP5 and reference strains (*S. glacialis* C16y^T^, *S. psychrolutea* MDB1-A^T^, and *S. paucimobilis* NCTC 11030^T^) were retrieved from GenBank with accession numbers GCA_004354345.1, GCA_014653575.1, GCF_014636175.1, and GCA_900457515.1, respectively. iDDH was performed between AAP5 and reference strains using the Genome-to-Genome Distance Calculator (GGDC 2.1) web server [41]. To support the iDDH results, ANI was calculated using the EzBiocloud web server [42]. The genomic G + C content of the AAP strain was taken from the published genome record (GCA_004354345.1). Orthologous gene cluster analysis was performed using the OrthoVenn2 web server [43]. Pairwise sequence similarities between all input protein sequences were calculated with an E-value cut-off of 1e^−2^. An inflation value *(-I)* of 1.5 was used to define orthologous cluster structure.

*Identification of DNA methylation sites.* PacBio-sequencing was performed as previously described [44]. Methylome analysis was performed using the “RS_Modification_and_Motif_Analysis.1” protocol included in SMRT Portal version 2.3.0. Only modifications with an identification phred score > 30 were considered.

*Physiological and biochemical characterization.* For physiological experiments with the AAP5 standard R2A liquid or solid medium (DSMZ medium 830) was used. Growth was monitored by measuring colony size on plates and optical density (λ = 650 nm) in the culture. Growth at 0, 0.5, 1, 5, 10, 50 and 70 g NaCl l^−1^ was examined in R2A liquid medium under dark conditions. The pH range for growth was investigated at pH 6–10 in increments of 1 pH unit, with an additional test at pH 7.3. The following pH buffer solutions were used: acetate buffer solutions (acetic acid/sodium acetate) for pH 6, NaH_2_PO_4_/Na_2_HPO_4_ for pH 6–8, Tris-HCl buffer for pH 9, and Tris buffer for pH 10. Cell motility was tested using motility assay as described previously [45]. Growth at 8, 23, 25, 27, and 37 °C was examined on R2A agar plates under dark conditions with an incubation time of 1 week. Catalase activity was determined by observing bubble production in a 3% H_2_O_2_ solution. Oxidase activity was determined by monitoring the oxidation of tetramethyl *p*-phenylenediamine dichloride on filter paper. Antibiotic susceptibility tests were performed using the disc diffusion method with commercially available discs (BioRad, CA, USA). Nutrient source utilization was assayed using Phenotype MicroArrays (BIOLOG, Inc., Hayward, CA, USA). The system was modified for use with an organic medium containing [L^−1^] 0.05 g glucose, 0.05 g peptone, 0.05 g yeast extract, 0.03 g sodium pyruvate, 0.3 g K_2_HPO_4_, and 1 g NaCl. Phenotype MicroArrays were incubated at 22 °C under aerobic and dark conditions. OD_750_ was measured using an Infinite F200 spectrophotometer (Tecan Trading AG, Mannendorf, Switzerland) after 1–4 days. 

*Other analyses.* For spectroscopic analyses, the cells harvested directly from agar plates were resuspended in 70% glycerol to reduce scattering. The in vivo absorption spectra were recorded using a Shimadzu UV 2600 spectrophotometer equipped with an integrating sphere. The same cells were also used to record fluorescence emission spectra. Fluorescence was excited by a single Cyan (505 nm) Luxeon Rebel light-emitting diode (Quadica Developments Inc., Canada). The emission spectra were recorded by a QEPro high-sensitivity fiber optics spectrometer (OceanOptics, FL, USA). Pigments were analyzed using high-performance liquid chromatography as described previously [23]. BChl *a* peaks were detected at 770 nm and its content was normalized on a per total protein basis. Lowry assay was used for protein extraction (Total Protein Kit, Micro Lowry, Peterson’s Modification, Sigma-Aldrich). Protein absorption was determined with UV-500 Thermo Scientific spectrophotometer at 650 nm. Respiratory quinones were extracted and analyzed as described previously [46].

Analyses of fatty acids and polar lipids were carried out by the Identification Service of the Deutsche Sammlung für Mikroorganismen und Zellkulturen (DSMZ), Germany. For this purpose, the strain was grown aerobically in full-strength R2A medium at 22 °C under 12-h dark/12-h light regime. Cells were harvested by centrifugation at 6000× *g* after reaching the late exponential phase (approx. OD_650_ = 0.8), freeze-dried, and sent to DSMZ.

## 3. Results and Discussion

*Cultivation and physiology.* The strain AAP5 forms yellow colonies on R2A agar. In liquid culture, growth of the strain occurred at a wide temperature range 8–37 °C, with an optimum temperature of 25–27 °C. The pH range for growth was 7.0–8.0, with an optimum at pH 7.0. Under optimal salinity (1 g NaCl l^−1^) the value for pH optimum shifted to 7.54. AAP5 did not require NaCl for growth, but it tolerated it up to 5 g NaCl l^−1^. The highest protein content was measured at 1 g NaCl l^−1^. Under optimal (1 g NaCl l^−1^) and higher than optimal salinity the cells of strain AAP5 were more abundant but smaller, which can be explained by a shorter division time. 

*Morphology.* Epifluorescence microscopy showed that AAP5 cells are rods with a clear IR autofluorescence signal of BChl *a* (Figure 1A left). The shape of the cells corresponds to the AAP morphotype C reported from the Gossenköllesee [12]. SEM images (Figure 1A center) confirmed the rod-like shape of AAP5 cells with length of 1.8 ± 0.3 μm in solitary cells and up to 3 μm in cells prior their dividing. TEM images (Figure 1A right) revealed a corrugated surface of a thin dense lipopolysaccharide layer above the outer membrane of the G-cell wall. The cell is additionally surrounded by fine mucilaginous sheaths of 140 ± 10 nm thickness resulting in an overall cell width of 0.8 ± 0.1 μm. AFM images documented that mean cell length was 2.9 ± 0.5 μm varying between 2.2–3.6 μm, occasionally forming short chains containing up to four cells. Two conjoined cells were usually longer with mean length of 4.6 ± 0.1 μm (min-max 4.4–4.7 μm). Mean cell width reported by AFM was 1.5 ± 0.2 μm, ranging from 1.2 to 1.8 μm. The wider cell dimensions apparent in AFM scans is due to the cell surface is coated by a soft diffusive mucilage (Figure 1B center) that was also seen in the TEM images. Map of Young’s modulus shows soft surface of YM = 710 ± 260 kPa. Several narrow stiffer areas (YM = 3.3 ± 1.5 MPa) orthogonal to the longer cell axis marking evidence of a site of a future cell division. The cell’s surface exhibits generally low adhesion of 95 ± 27 pN (Figure 1B right). RMS topography roughness was low over most of the cell surface (20–30 nm) but consistently higher above the division regions and at the cell terminus (50–70 nm).

*Physiological and biochemical characteristics.* Strain AAP5 and *S. glacialis* C16y^T^ could be differentiated by cell properties, NaCl concentration tolerance, temperature range and optimum for growth, utilization of carbon sources, as well as susceptibility to antibiotics (Table 1). Strain AAP5 was able to utilize the following compounds as a carbon source: pyruvic acid, succinic acid, L-malic acid, citramalic acid, α-keto-butyric acid, β-hydroxy butyric acid, D-tartaric acid, D-glucuronic acid, D-galacturonic acid, L-arabinose, D-glucose, D-galactose, maltose, D-mannose, D-melibiose, L-rhamnose, glycerol, sucrose, lactulose, uridine, L-glutamine, α/β/γ-cyclodextrin, dextrin, pectin, and amygdalin. It was not able to utilize citric acid, acetic acid, D-gluconic acid, adipic acid, capric acid, phenylacetic acid, D-sorbitol, D-mannitol, and inositol. The ability to utilize ammonia, N-amylamine, N-butylamine, ethylamine, ethanolamine, putrescine, β-phenylethylamine, acetamine, glucuronamide, N-acetyl-D-glucosamine, N-acetyl-D-mannosamine, cytidine, guanosine, uridine, xanthine, ε-amino-N-caproic acid, and δ-amino-N-valeric acid as a nitrogen source was also observed. From 22 proteinogenic amino acids, only L-aspartic acid, L-glutamic acid, L-serine, L-tyrosine, L-valine, and D/L-glutamic acid were utilized. It did not utilize nitrite, nitrate, and urea. 

AAP5 exhibited natural resistance to (μg ml^−1^) ciprofloxacin (5), erythromycin (15), neomycin (6), ofloxacin (5), and tetracycline (30), but it was sensitive to cefoxitin (30), gentamicin (10), and penicillin G (100). The major respiratory quinone was ubiquinone-10. The predominant (>10%) fatty acid was C_18:1_*ω*7*c* (64.6%). The polar lipid profile of AAP5 contained phosphatidylglycerol, diphosphatidylglycerol, phosphatidylethanolamine, sphingoglycolipid, six unidentified glycolipids, one unidentified phospholipid, and two unidentified lipids. The fatty acid and polar lipid profiles agreed with those of *S. glacialis* C16y^T^ [47]. The cells were positive for oxidase and catalase.

The main carotenoid was nostoxanthin. The in vivo absorption spectrum of AAP5 cells displayed a clear single near IR BChl *a* absorption band at 872 nm. In the blue part of the spectrum there was an intense absorption of carotenoids, with main absorption peaks at 433, 458, and 489 nm (Figure 2). As reported previously BChl *a* was detected in cells grown on R2A agar plates, but not in cells grown in full-strength R2A broth [23]. 

Although containing genes (E2E30_RS14635–E2E30_14725) necessary for flagellar biosynthesis and assembly, under the experimental conditions we used AAP5 was non-motile. Moreover, no structures similar to the flagella were found by neither of the microscopic techniques use to visualize the cells.

*Phylogenomy and genomic traits.* AAP5 genome contains three copies of the 16S rRNA gene. Two copies are identical (at position 3117387..3118878 and 3704698..3706189), the third one (at position 905790..907281) differs in one nucleotide position. For the alignment we used sequence from the two identical copies. The 16S rRNA tree (Figure 3) showed that AAP5 strain grouped with the genus *Sphingomonas* and formed a distinct cluster with *S. glacialis* (98.3% pairwise 16S rRNA sequence similarity), *S. psychrolutea* (96.8%), and *S. melonis* (96.5%). Further evidence provided *in silico* comparison of genomic sequences of strain AAP5 and reference strains of three species, *S. glacialis* C16y^T^, *S. psychrolutea* MDB1-A^T^, and the type species of the genus *S. paucimobilis* NCTC 11030^T^. In silico DNA-DNA hybridization values between strain AAP5 and *S. glacialis* C16y^T^, between strain AAP5 and *S. psychrolutea* MDB1-A^T^, and between strain AAP5 and *S. paucimobilis* NCTC 11030^T^ were 74.50 ± 2.95%, 23.60 ± 2.4%, and 20.0 ± 2.3%, respectively. The cut-off value for species delineation is 70%. Average nucleotide identity values, representing mean identity values between a given pair of genomes, were 97.09% between strain AAP5 and *S. glacialis* C16y^T^, 80.35% between strain AAP5 and *S. psychrolutea* MDB1-A^T^, and 74.26% between strain AAP5 and *S. paucimobilis* NCTC 11030^T^. The proposed boundary for defining a novel species is 95–96% [50]. In addition, we calculated the numbers of shared orthologous gene clusters among strain AAP5 and the two reference strains. Strain AAP5 shares 2,174 orthologous gene clusters with the type strain of *S. pauci-mobilis* NCTC 11030T. The number of unique orthologous gene clusters (620) shared by AAP5 and *S. glacialis* C16y^T^ was much greater than the number shared by either AAP5 and *S. psychrolutea* MDB1-A^T^ (44) or AAP5 and *S. paucimobilis* NCTC 11030^T^ (85), indicating a close relationship between AAP5 and *S. glacialis* C16y^T^ also at the genomic level. In summary, phylogenetic and genomic analyses imply that AAP5 belongs to the same genus as the three reference *Sphingomonas* strains with *S. glacialis* C16y^T^ representing its closest relative. The genome of AAP5 consists of four replicons (Figure 4) including one circular chromosome (3,987,367 bp; GC content 66.2%), and three plasmids p32 (31,551 bp; GC content 63.3%), p150 (150,273 bp; GC content 63.3%), and p213 (213,080 bp; GC content 62.7%) with a total length of 4,382,271 bp encoding 4,128 genes. Overall, GC content is 65.9%. Similar GC content of all four replicons suggests their common origin. The chromosome contains three copies of rRNA operons (5S, 16S, 23S), 51 tRNA genes, and 4,065 protein-coding sequences. It is notable that the AAP5 chromosome contains one continuous 38.6-kb-long photosynthesis gene cluster (E2E30_16220–E2E30_16405) and a gene coding for xanthorhodopsin (E2E30_05030), a transmembrane protein with (presumably) proton-pumping activity [22]. Furthermore, AAP5 contains a complete and contiguous gene cluster (E2E30_17610–E2E30_17700) with genes coding for gene transfer agents (GTA), virus-like particles transferring pieces of genomic DNA between prokaryotic cells. The gene organization is similar to the GTA cluster in *Rhodobacter capsulatus* [51]. The *pufL* and *pufM* genes, coding proteins of the type 2 reaction centers, represent standard genetic markers of anoxygenic phototrophs and were used to infer the phylogenetic relationship of strain AAP5 with other phototrophic *Proteobacteria*. The ML phylogenetic tree shows a split between basic alphaproteobacterial orders (Rhodobacterales, Sphingomonadales, Rhizobiales) (Figure 5). The AAP5 strain clearly clusters with other photosynthetic *Sphingomonas*. Similarly, as in the 16S rRNA tree, *Sphingomonas glacialis* C16y^T^ represents its closest relative. Interestingly, inside of the Sphingomonadales branch lays also one species from a distinct order, *Polynucleobacter duraquae* DSM 21495^T^ (Burkholderiales) (Figure 5). The presented tree agrees well with recently published PufLM phylogeny of phototrophic Proteobacteria [52,53]. 

*DNA methylation.* Single-molecule real-time sequencing allows for detection of any DNA modifications [54]. Three different methylation sites are commonly found in bacteria [55]. We identified 19502 N^6^-Adenosine (m6A) and 420 N^4^-Cytosine (m4C) bases but no N^5^-Cytosine (m5C) motifs on the chromosome and plasmids of AAP5 (Figure 6). The only present m4C-motif was restricted to the chromosome and plasmid p213, whereas the five m6A motifs were found on all four replicons. By far, the most abundant was the GANTC motif that is m6A-methylated by the methyltransferase CcrM [56]. This enzyme is highly conserved in Alphaproteobacteria. The GANTC motif is overrepresented in intergenic regions and underrepresented in genes [57]. In *Caulobacter vibrioides* CcrM-activity is restricted to the pre-divisional cell, thus newly replicated DNA stays hemimethylated until replication has finished. The activity of several *Caulobacter vibrioides* promoters is dependent on the methylation state and, thereby, coupled to replication [58,59]. The density of GANTC-motifs is similar for the chromosome and plasmid p32 but lower for the plasmids p213 and p150. The other five m6A-motifs as well as the m4C motif might represent targets for restriction-modification systems [60]. The motif CATGAG is underrepresented on the chromosome compared to the plasmids and might, therefore, have a regulatory function for extrachromosomal elements. 

## 4. Conclusions

On the basis of phylogenetic and genomic evidence, strain AAP5 belongs to the species *S. glacialis*. This organism contains genes for anoxygenic photosynthesis as well as xanthorhodopsin. Although phototrophy is common among Alphaproteobacteria, the common presence of two phototrophic systems in *Sphingomonas* represents a unique phenomenon, which deserves further attention.

## Figures and Tables

**Figure 1 microorganisms-09-00768-f001:**
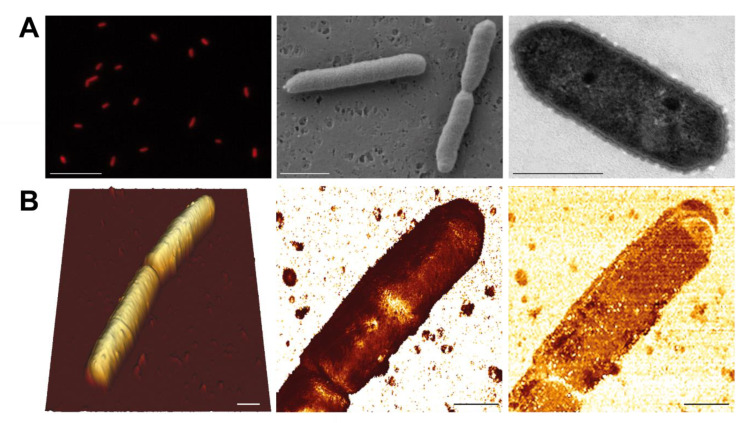
Microscopy images of the studied AAP5 strain. (**A**, Left) Infrared epifluorescence microscopy image (false color) showing autofluorescence from BChl *a*. *(Center)* Scanning electron microscopy image. (Right) Transmission electron microscopy image. Scale bars represent 10 μm, 1 μm, and 0.5 μm, respectively. (**B**) Atomic force microscopy images of AAP5 cells. (Left) 3-D topography image of two conjoined cells. Full image vertical range is 1.5 um. (Center) Map of Young’s modulus of the upper cell shown on right reveals elastic surface (YM = 710 ± 260 kPa) with a narrow stiffer area (YM = 3.3 ± 1.5 MPa) orthogonal to the longer cell axis. Full vertical range of the image is 4 GPa. (Right) Cell surface is decorated by a soft diffusive mucilage responsible for the low stiffness and generally low adhesion of 95 ± 27 pN. Full image vertical range is 300 pN. Cell terminus exhibits both higher roughness (50–70 nm) and adhesion (156 ± 20 pN) than the rest of the cell. Scale bars represent 1 μm.

**Figure 2 microorganisms-09-00768-f002:**
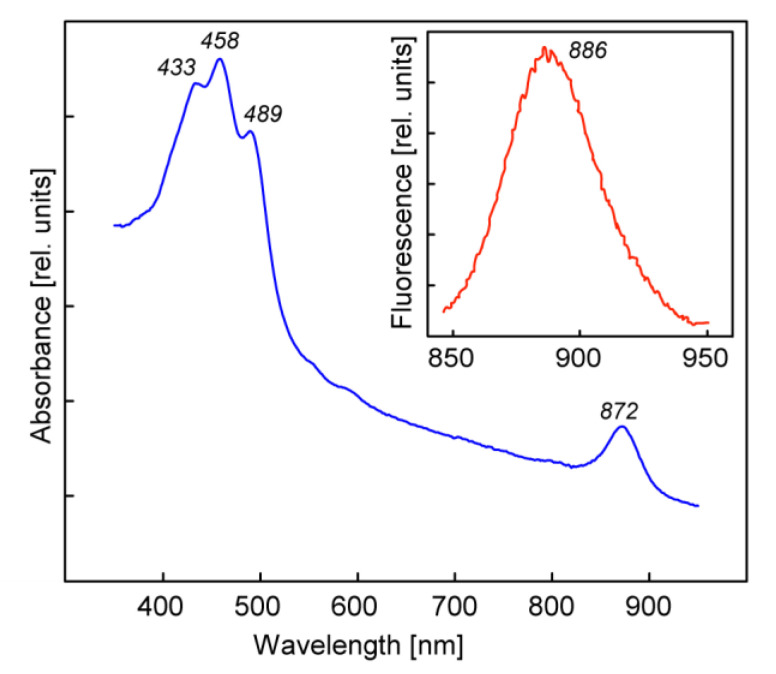
In vivo absorption spectra of *Sphingomonas* sp. AAP5 cells. *(Inset)* The fluorescence emission spectrum.

**Figure 3 microorganisms-09-00768-f003:**
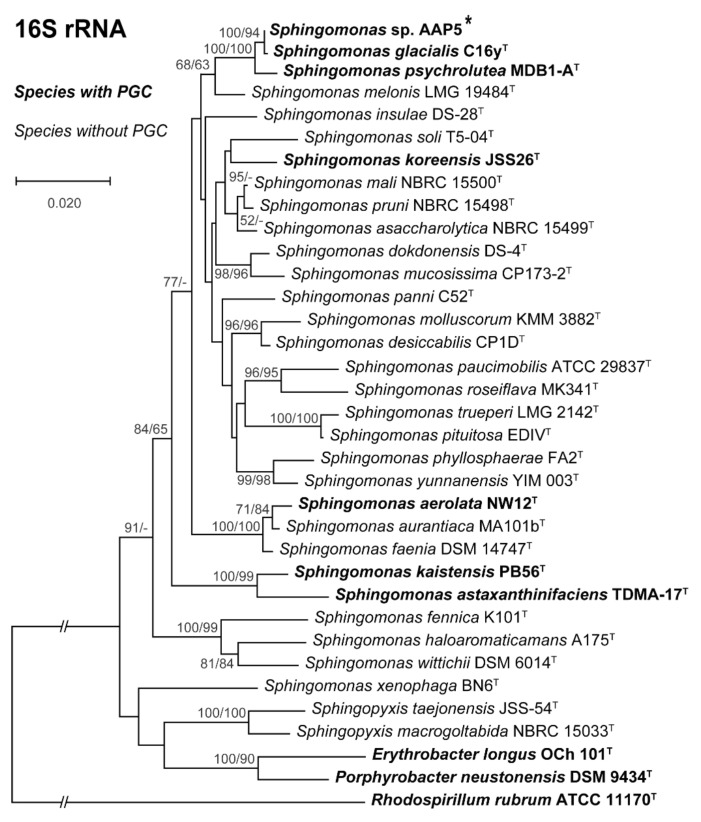
16S rRNA phylogenetic tree showing position of the AAP5 strain (marked by the asterisk) within the genus *Sphingomonas* and some representatives of other related taxa. Strains with photosynthesis gene cluster (PGC) are in bold. Phylogenetic tree was based on 16S rRNA gene sequences downloaded from the SILVA database and NCBI GenBank (June, 2020). Nucleotide sequences were aligned using ClustalW resulting in alignment with 1,313 conserved nucleotide positions (after ambiguously aligned regions and gaps being manually excluded). The phylogenetic tree was calculated using both neighbor-joining (NJ) and maximum likelihood (ML) algorithms and 1,500× bootstrap replicates. *Rhodospirillum rubrum* ATCC 11170^T^ was used as an outgroup organism. Scale bar represents changes per position. NJ/ML bootstrap values >50% are shown.

**Figure 4 microorganisms-09-00768-f004:**
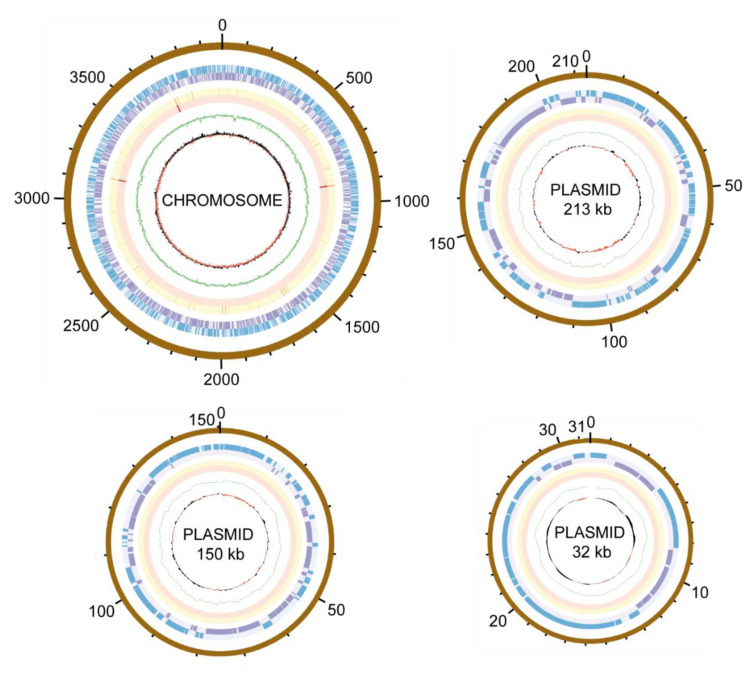
Circular representations of the AAP5 chromosome and three plasmids. The outer to inner rings represent: scale of replicon size in kb; position of open reading-frames encoded on the plus/minus strand (in blue/purple); tRNA (orange); rRNA (red); GC content (in green); GC skew (in red/black).

**Figure 5 microorganisms-09-00768-f005:**
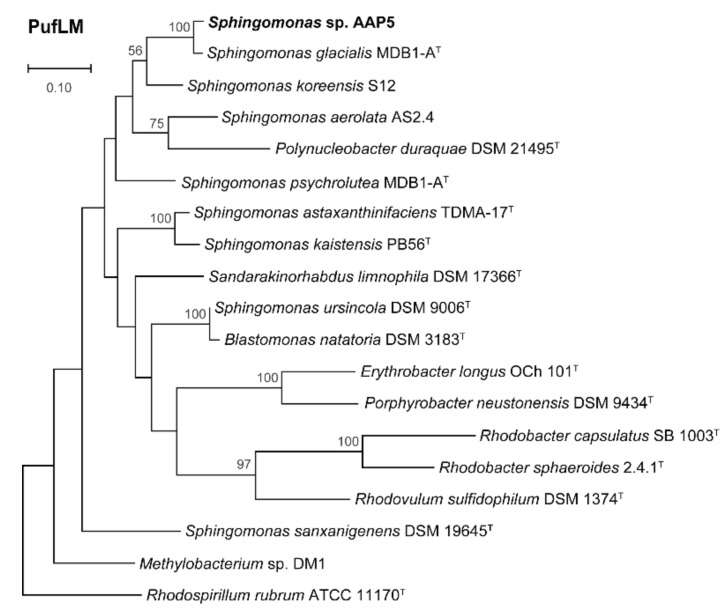
Maximum likelihood (ML) phylogenetic tree based on concatenated alignments of amino acid sequences of the photosynthetic reaction center subunit L and M (PufLM; 573 common amino acid positions). The ML tree was calculated using LG model and bootstrap 1,000x. *Rhodospirillum rubrum* ATCC 11170^T^ was used as an outgroup organism. Scale bars represent changes per position. Bootstrap values >50% are shown. Studied strain is in bold.

**Figure 6 microorganisms-09-00768-f006:**
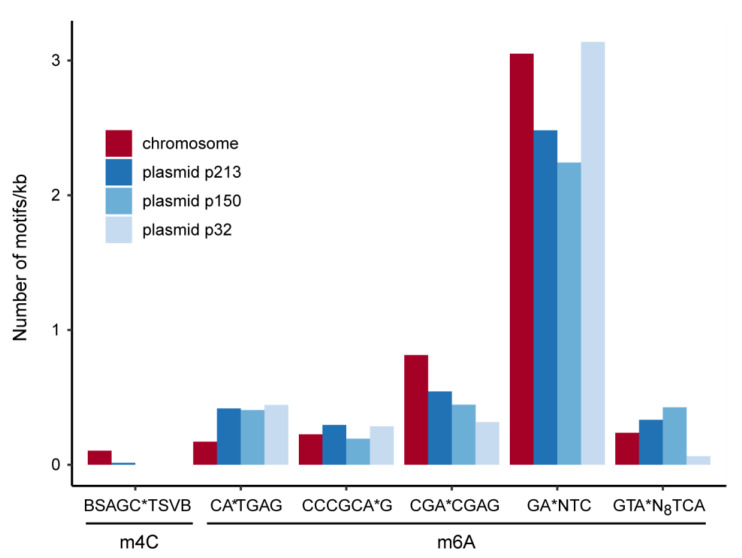
DNA methylation motifs inferred from SMART sequencing. The number of motifs has been normalized to one kb sequence length. The methylation site is indicated by an asterisk.

**Table 1 microorganisms-09-00768-t001:** The comparison of strain AAP with other three *Sphingomonas* strains: 1, *Sphingomonas* sp. AAP5 (data were taken from this study); 2, *Sphingomonas glacialis* C16y^T^ [47,48]; 3, *Sphingomonas psychrolutea* MDB1-A^T^ [48]; 4, *Sphingomonas melonis* LMG 19484^T^ [14]; 5, *Sphingomonas paucimobilis* NCTC 11030^T^ [1,49].

Parameter	1	2	3	4	5
Colony color	Yellow	Yellow	Orange-yellow	Deep yellow	Yellow
Cell width [μm]	0.7–0.8	0.5	0.5–0.6	0.68–0.85	0.7
Cell length [μm]	1.1–2.3	0.8	1.8–2.2	1.2–1.9	1.4
Motility	No	No	No	No	Yes
Genome characteristics ^‡^					
G + C content [%]	65.9	65.7	64.2	67.1	65.7
PGC	Yes	Yes	Yes	No	No
Xanthorhodopsin	Yes	Yes	Yes	No	No
Utilization of					
D-mannose	Yes	No	Yes	Yes	No
D-melibiose	Yes	No	No	No	Yes
L-rhamnose	Yes	No	No	No	No
Phenylacetate	No	No	No	Yes	No
Antibiotics resistance					
Penicillin G 100 μg ml^−1^	No	Yes	*n.d.*	Yes	Yes
Tetracycline 30 μg ml^−1^	Yes	No	*n.d.*	No	No

^‡^ Based on published genome sequences (accession numbers GCA_004354345.1, GCA_014653575.1, GCA_014636175.1, GCA_001761345.1, and GCA_900457515.1, respectively). *n.d.*, no data.

## Data Availability

The complete genome sequence is deposited at NCBI GenBank under the accession numbers: CP037913 (chromosome) and CP037914-CP037916 (plasmids). The GenBank accesion number for the 16S rRNA sequence is MW410774.

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
