# Peer review of "Characterization of the Aerobic Anoxygenic Phototrophic Bacterium *Sphingomonas* sp. AAP5"

_microorganisms, 2021, doi:10.3390/microorganisms9040768_

Round 1

Reviewer 1 Report

Due to the similarity between S. glacialis, S. psychrolutea and AAP5, the authors need include in the comparisons S. psychrolutea, in the discussion and in table 1 and Figs. 3 and 5.

Other comments are included in the attached document.

Author Response

Due to the similarity between S. glacialis, S. psychrolutea and AAP5, the authors need include in the comparisonsS. psychrolutea, in the discussion and in table 1 and Figs. 3 and 5.

We included S. psychroluteaMDB1-ATinto comparisons (page 3, line 133 and 135), RESULTS AND DISCUSSION section (page 7, line 275,277, 279, 280, 284, 291), Table 1, Figure 3, and Figure 5.

Responses to the comments included in the attached document:

Page 1 and 2, line 43,44: We added the reference Takeuchi et al., 2001 as suggested by the Reviewer 1 into the INTRODUCTION section.Page 2, line 61: We deleted the “S.“ abbreviation.

What was the result of the motility assay?

Result of the motility assay was negative (page 6, 24, line 255).

Check scale bar for the Fig. 1A, center. According to the scale bar, the cells must measure 1.2 –2.8 μm in length.

We re-checked the scale bar. Cells in the Fig. 1A, center are longer than of 1.8±0.3 μm becauseof their imminent cell division. We clarified this issue on page 4, line 197, 198.

We corrected Table 1 as suggested by the Reviewer 1.

Reviewer 2 Report

The manuscript is very well written, the data is well represented.
Please review the article for grammatical errors which are present in quite a few paras. Please check the legend for figure 2.

Author Response

The manuscript is very well written, the data is well represented.
We thank the reviewer for his/her positive evaluation of our work. 

Please review the article for grammatical errors which are present in quite a few paras. 
We double-checked the manuscript for grammatical errors and typos. 

Please check the legend for Figure 2.
We deleted redundant text from the legend for Figure